# Efficacy of colchicine in patients with moderate COVID-19: A double-blinded, randomized, placebo-controlled trial

Motlabur Rahman[1], Ponkaj K. Datta[1]*, Khairul Islam[1], Mahfuzul Haque[1], Reaz Mahmud[2], Uzzwal Mallik[1], Pratyay Hasan[1], Manjurul Haque[1], Imtiaz Faruq[1], Mohiuddin Sharif[1], Rifat H. Ratul[1], Khan Abul Kalam Azad[1¤a], Titu Miah[3], Md. Mujibur Rahman[1¤b]

1 Department of Medicine, Dhaka Medical College, Dhaka, Bangladesh, 2 Department of Neurology, Dhaka Medical College, Dhaka, Bangladesh, 3 Department of Medicine and Principal, Dhaka Medical College, Dhaka, Bangladesh

¤a Current address: Department of Medicine, Popular Medical College, Dhaka, Bangladesh
¤b Current address: Faculty of medicine, Bangabandhu Sheikh Mujib Medical University, Dhaka, Bangladesh
* ponkajdatta@yahoo.com

**Data Availability Statement:** All data set files are available from the https://figshare.com/articles/dataset/Colchicine_in_COVID-19_BD_trial_data_

## Abstract

### Background

Severe acute respiratory syndrome coronavirus 2 (SARS-CoV-2) infection may cause severe life-threatening diseases called acute respiratory distress syndrome (ARDS) owing to cytokine storms. The mortality rate of COVID-19-related ARDS is as high as 40% to 50%. However, effective treatment for the extensive release of acute inflammatory mediators induced by hyperactive and inappropriate immune responses is very limited. Many anti-inflammatory drugs with variable efficacies have been investigated. Colchicine inhibits inter-leukin 1 beta (IL-1β) and its subsequent inflammatory cascade by primarily blocking pyrin and nucleotide-binding domain leucine-rich repeat and pyrin domain containing receptor 3 (NLRP3) activation. Therefore, this cheap, widely available, oral drug might provide an added benefit in combating the cytokine storm in COVID-19. Here, we sought to determine whether adding colchicine to other standards of care could be beneficial for moderate COVID-19 pneumonia in terms of the requirement for advanced respiratory support and mortality.

### Methods and findings

This blinded placebo-controlled drug trial was conducted at the Dhaka Medical College Hospital, Dhaka, Bangladesh. A total of 300 patients with moderate COVID-19 based on a positive RT-PCR result were enrolled based on strict selection criteria from June 2020 to November 2020. Patients were randomly assigned to either treatment group in a 1:1 ratio. Patients were administered 1.2 mg of colchicine on day 1 followed by daily treatment with 0.6 mg of colchicine for 13 days or placebo along with the standard of care. The primary outcome was the time to clinical deterioration from randomization to two or more points on a seven-category ordinal scale within the 14 days post-randomization. Clinical outcomes

set/16734640 database DOI: 10.6084/m9.figshare.
16734640.

**Funding:** The author(s) received no specific
funding for this work.

**Competing interests:** The authors have declared
that no competing interests exist.

were also recorded on day 28. The primary endpoint was met by 9 (6.2%) patients in the placebo group and 4 (2.7%) patients in the colchicine group (P = 0.171), which corresponds to a hazard ratio (95% CI) of 0.44 (0.13–1.43). Additional analysis of the outcomes on day 28 revealed significantly lower clinical deterioration (defined as a decrease by two or more points) in the colchicine group, with a hazard ratio [95%CI] of 0.29 [0.098–0.917], (P = 0.035). Despite a 56% reduction in the need for mechanical ventilation and death with colchicine treatment on day 14, the reduction was not statistically significant. On day 28, colchicine significantly reduced clinical deterioration measured as the need for mechanical ventilation and all-cause mortality.

## Conclusion

Colchicine was not found to have a significant beneficial effect on reducing mortality and the need for mechanical ventilation. However, a delayed beneficial effect was observed. Therefore, further studies should be conducted to evaluate the late benefits of colchicine.

## Clinical trial registration

**Clinical trial registration no:** ClinicalTrials.gov Identifier: NCT04527562 https://www.google.com/search?client=firefox-b-d&q=NCT04527562.

## Introduction

The World Health Organization (WHO) formally notified the pneumonia case cluster in Wuhan City, China on December 31, 2019 [1]. By mid-January 2020, the virus and its genome sequence were identified. The novel coronavirus, severe acute respiratory syndrome coronavirus 2 (SARS-CoV-2), was isolated and the disease was later named coronavirus disease-19 (COVID- 19) [2].

SARS-CoV-2 infection may cause an asymptomatic, mild-to-severe, life-threatening disease [3]. Death is mainly caused by acute respiratory distress syndrome (ARDS), which involves a cytokine storm. The mortality rate of COVID-19-related ARDS is between 40% and 50% [4].

The cytokine storm syndrome indicates a hyperactive and inappropriate immune response characterized by the release of widespread acute inflammatory mediators, such as interferon, interleukins, tumor necrosis factors, and chemokines [5]. This syndrome can also occur in a patient with a diminishing viral load, which indicates that it may be caused by an exuberant host immune response [6]. Effective treatments to reduce this hyperactive immune response are limited. However, many anti-inflammatory drugs with variable efficacy have been investigated, including targeted anti-inflammatory therapy, such as interleukin 1 (IL 1) inhibitors, IL 6 inhibitors, interferon-gamma, tumor necrosis factor alpha (TNF-α) inhibitors, and non-targeted anti-inflammatory agents, such as corticosteroids, hydroxychloroquine, Janus kinase (JAK) inhibitors, and colchicine. Among them, corticosteroids, such as dexamethasone, have been found to have beneficial effects in hospitalized patients requiring supplemental oxygen [7]. Colchicine has long been used as an anti-inflammatory agent for Bechet's disease, Lepra reaction, gout, and familial Mediterranean fever. Colchicine inhibits IL-1β and its subsequent inflammatory cascade by primarily blocking pyrin and nucleotide-binding domain leucine-rich repeat and pyrin domain containing receptor 3 (NLRP3) activation. As NLRP3 is likely activated following viral entry into cells, colchicine might provide an added benefit in combating the cytokine storm in COVID-19. Colchicine is also inexpensive, widely available, and can

be administered orally. Further, its side effect profile is well known and limited. Therefore, colchicine has a widespread advantage over other costly and parenteral anti-inflammatory drugs, such as tocilizumab and anakinra [8]. One case series, retrospective observational studies, and randomized control trials revealed variable efficacies of colchicine in COVID-19 [9–13]. Among several systematic reviews on COVID-19 and colchicine, the most recent COCHRANE review is identified as a comprehensive and rigorous assessment. This review included three randomized controlled trials with 11,525 hospitalized participants and one randomized controlled trial with 4,488 non-hospitalized participants. According to the study, colchicine might have little to no influence on mortality or clinical progression in hospitalized patients with moderate-to-severe COVID-19. However, there is uncertain evidence of the effect of colchicine on mortality in people with asymptomatic infection or mild disease. Further, colchicine might cause a slight reduction in hospital admissions or deaths within 28 days and the rate of serious adverse events compared with placebo [14]. Here, we sought to determine whether adding colchicine to other standard of care treatments could be beneficial for patients with moderate COVID-19 pneumonia in terms of the requirement for advanced respiratory support and mortality.

## Patients and methods

### Ethics approval

The trial was conducted in accordance with the principles of the Good Clinical Practice Guidelines of the International Conference on Harmonization-2016 and was approved by the ethical review committee of Dhaka Medical College Hospital (ERC-DMC/ECC/2020/128, dated June 08, 2020). This trial was registered at ClinicalTrials.gov (identifier: NCT04527562). The trial protocol was published, and the full text is available at https://dx.doi.org/10.18203/2349-3259. ijct2021xxxx (Rahman et al). Written informed consent was obtained from all patients. The study was conducted in accordance with the Equator Network Guidelines.

This was an investigator-initiated, blinded, and placebo-controlled trial. Treatment safety and efficacy were monitored by an independent data safety monitoring board at the Dhaka Medical College. Routine investigations were performed in the pathology laboratory of the Dhaka Medical College. Polymerase chain reaction (PCR) tests were performed, free of charge, in the virology laboratory of the Dhaka Medical College. Colchicine and placebo were supplied by Incepta Pharmaceuticals Ltd. Shahid Tajuddin Ahmed Srani, Tejgaon industrial area, Dhaka, Bangladesh. A random number was generated and maintained by an independent biostatistician from Incepta Pharmaceuticals Limited, Dhaka, Bangladesh. The company had no role in the planning, data collection, analysis, or interpretation of the study results.

### Patients

The study was conducted at Dhaka Medical College Hospital, Dhaka, Bangladesh. A total of 300 patients were enrolled between June 2020 and November 2020. These patients had moderate COVID-19 confirmed by positive RT-PCR (within 3 days of positivity) results and were older than 18 years; both sexes were included in the trial. In this study, moderate disease was defined clinically as fever or history of fever, cough and/or shortness of breath, respiratory rate <30 breaths/minute, oxygen saturation 94% or more without any supplemental oxygen, and pulmonary consolidations involving less than 50% of lungs based on chest imaging (chest x ray or CT scan of chest) (i.e., all patients in both arms had radiologically confirmed pneumonia at enrolment). Pregnant and lactating mothers; patients with known hypersensitivity to colchicine; known chronic illnesses such as hepatic failure, chronic kidney disease (eGFR< 30 ml/ min), decompensated heart failure, long QT syndrome (QTc >450 msec), inflammatory

bowel disease, chronic diarrhea, or malabsorption; patients taking colchicine for other indications (mainly chronic indications represented by familial Mediterranean fever or gout); and patients undergoing chemotherapy for cancer were excluded from the study. All patients enrolled in this study were assigned to category 3, were hospitalized, and did not require supplemental oxygen, according to a seven-category ordinal scale (as defined in the Outcome Measures section). Patients in other categories were excluded from the study.

## Trial design

**Random assignment.** On day 1 of enrolment, patients were randomly assigned to either of the treatment groups at a 1:1 ratio. A random number was generated and maintained by an independent biostatistician from Incepta Pharmaceuticals Limited, Dhaka, Bangladesh. Group assignment was concealed using an identical opaque envelope. Colchicine and placebo blisters within each envelope were identical in size and labelling. A total of 300 identical cards numbered 1–300 were prepared. After signing the informed consent form, each participant randomly took a card from the investigator. The study nurse then supplied a sealed envelope with the corresponding number. Neither the investigators nor the patients were aware of the group assignment.

**Details of blinding in this randomized trial of colchicine for the treatment of adult patients with moderate COVID-19**

| Individual blinded | Information withheld | Method of blinding | Blinding maintained |
|---|---|---|---|
| Person assigning participants to groups | Group assignment | opaque envelops | Yes |
| Participants | Group assignment | Placebo medications | Yes |
| Care providers | Group assignment | Placebo medications | Yes |
| Data collectors and managers | Group assignment | Not informed of group assignment | Yes |

Decoding was performed at the end of the trial under the supervision of the Data Safety Monitoring Board of the Dhaka Medical College, Dhaka, Bangladesh.

## Interventions

**Active drug group.** The treatment group received a starting dose of 1.2 mg of colchicine (2 tablets of 0.6 mg) 12 h divided doses on day 1 followed by daily treatment with 0.6 mg of colchicine for 13 days. Paracetamol, antihistamines, and oxygen therapy were also administered as part of the standard care according to the National Guidelines of Bangladesh and Clinical Management of COVID-19, Interim Guidance of World Health Organization [15,16]. Low molecular weight heparin, according to the indication, appropriate broad-spectrum antibiotics, if needed, remdesivir injection, and other drugs for associated comorbid conditions were prescribed by the attending physicians.

**Placebo group.** The placebo group received standard care and placebo tablets that appeared similar to the study drugs (i.e., the placebo resembled a colchicine tablet). Colchicine has a bitter taste. As the placebo tablets only contained excipients, the bitter taste could not be replicated in the placebo.

## Experimental procedures

The baseline demographic and clinical characteristics of patients were collected using data collectors in a case-record form. The date of random assignment was considered day 1, and all patients received their initial treatment dose on day 1. Patients were followed up from day 1 through day 28 until discharge from the hospital due to clinical recovery and at the COVID-19

clinic of the hospital. Patients who failed to attend the clinic were followed up via telephone calls. Clinical recovery was defined as a normal body temperature of 36.1°C–37.2°C maintained for at least 3 days, significantly improved respiratory symptoms (respiratory rate < 25/ min, no dyspnea), and oxygen saturation greater than 93% without supplemental oxygen inhalation, as recommended by the national guideline of Bangladesh and the World Health Organization [15,16]. The clinical status and vital signs (including respiratory status) were recorded daily. Any adverse events defined by the Medical Dictionary for Regulatory Activities (MedDRA) were documented.

Laboratory tests were performed on day 1 and included the following: complete blood count; concentrations of random blood glucose, creatinine, alanine transaminase, C-reactive protein, serum ferritin, D-dimer, lactate dehydrogenase, and either a chest radiograph or chest CT scan, whichever was feasible or needed. Complete blood count was repeated on days 3, 7, 10, and 14, and random blood glucose, serum creatinine, and serum alanine transaminase tests were repeated on days 7 and 14. Real-time polymerase chain reaction for COVID-19 was scheduled to be performed at least twice; however, due to changes in national policy and resource constraints, only one run was performed for each patient during their hospital stay (i.e., after the initial positive test).

## Outcome measures

The primary outcome was the time to clinical deterioration, defined as the time from randomization to a change of two or more points (from the status at randomization) on a seven-category ordinal scale. The scale was recommended by the WHO R&D Blueprint expert group. The seven-category ordinal scale consisted of the following categories: 1) not hospitalized with the resumption of normal activities; 2) not hospitalized, but unable to resume normal activities; 3) hospitalized, not requiring supplemental oxygen; 4) hospitalized, requiring supplemental oxygen; 5) hospitalized, requiring nasal high-flow oxygen therapy or noninvasive mechanical ventilation, or both; 6) hospitalized, requiring ECMO or invasive mechanical ventilation, or both; and 7) death [17–19]. The secondary outcomes were: 1) length of hospital stay defined by the total number of days from enrolment to discharge from hospital or death in hospital; 2) proportion of participants requiring supplemental oxygen (any amount or by any device such as nasal cannula, face mask, high-flow nasal cannula, invasive or non-invasive mechanical ventilation); 3) proportion of participants requiring mechanical ventilation; and 4) proportion of all-cause mortality among the participants. The timeframe for all outcomes was 14 days post-randomization. Clinical outcomes on day 28 were also recorded.

## Statistical methods

The estimated sample size was 292 (146 in each arm). The following formula [20] was used to calculate the sample size:

$$n = \frac{(Z_{\alpha/2} + Z_\beta)^2 X P_1(1-P_1) + P_2(1-P_2)}{(P_1 - P_2)}$$

where n = sample size required in each group, $P_1$ = proportion of participants with a change in the placebo group at day 14 = 0.50, $P_2$ = proportion of participants with a change in the colchicine group at day 14 = 0.34, $P_1$-$P_2$ = clinically significant difference = 0.16, $Z_{\alpha/2}$: this is dependent on the level of significance; 5% = 1.96, $Z_\beta$: this is dependent on power; 80% = 0.84.

Due to the lack of data at the time of study initiation, we assumed that 50% of the hospitalized patients will have deteriorated with standard treatment plus placebo, and a difference of 16% between the colchicine and placebo would be sufficient. Therefore, a sample size of 292

patients, 146 in each arm, was sufficient to detect a clinically important difference of 16% between groups in the prevention of deterioration in moderate COVID-19 patients using a two-tailed Z-test of proportions between two groups, with 80% power and a 5% level of significance. Later, a study that reported 29% deterioration among hospitalized patients was found [21]. Using the same difference in efficacy, power, and level of significance, we obtained a sample size of 202, with 101 patients in each arm. In addition, we assumed a higher loss to follow-up rate during this pandemic; therefore, a total of 300 patients were recruited, 150 patients in the colchicine treatment group and 150 patients in the placebo group.

Primary efficacy analysis was performed on an intention-to-treat basis. Continuous parameters are reported as median and interquartile range (IQR) and were compared using the non-parametric test (Mann-Whitney U). For comparisons of the biochemical laboratory test values on days 1, 7, and 14, the Wilcoxon signed rank test was used. Categorical variables are reported as counts and percentages and were compared using the $\chi 2$ test. A Cox proportional hazards model was used for the final outcome analysis. A survival analysis for clinical deterioration was performed. The Kaplan-Meier curve and log-rank tests were performed to determine significance. Statistical significance was set at P = < 0.05; all tests were 2-tailed. IBM SPSS statistical software (version 23.0) was used for all statistical analyses. Subgroup analysis was performed for the different sexes, age subgroups (18–40 years, 41–60 years, and > 60 years), comorbidities (diabetes mellitus, hypertension, asthma, and COPD), and duration of symptoms before enrolment ($\leq$ 10 days vs. > 10 days). Odds ratios with a 95% confidence interval for the clinical endpoint were measured and a forest plot was constructed to identify any differences in the primary outcomes among the different subgroups. Additional analysis was performed using day 28 outcomes with the same statistical methods used for the day 14 outcomes.

## Results

Study enrolment began on July 14, 2020, and was completed on November 15, 2020. The last follow-up date was December 15, 2020. A total of 366 patients were screened for eligibility and 299 patients were included for randomization (Fig 1). After group allocation, 1 patient in the colchicine group and 2 patients in the placebo group withdrew their consent and did not receive the drug. Thus, 296 participants received the allocated treatment. The median [IQR] age of the participants was 47 [35–55] years. The baseline clinical characteristics and baseline blood biomarkers (Table 1) of the patients administered the allocated treatment were analyzed. The colchicine and placebo groups were similar in terms of demographic characteristics, clinical status, and laboratory evaluations at baseline. The baseline clinical score was 3 on a seven-category ordinal scale in both groups. All patients received additional treatment according to the national guidelines of Bangladesh and the WHO guidelines. An analysis was performed for low-molecular-weight heparin and dexamethasone among the treatment groups; however, no significant difference was found.

### Outcomes

The primary endpoint was met by nine (6.2%) patients in the placebo group and four (2.7%) patients in the colchicine group (P = 0.171), which corresponds to a hazard ratio (95% CI) of 0.44 (0.13–1.43) (Table 2). The median (IQR) time of a 2-point deterioration on the seven-category ordinal scale was 3 (2.2–10.5) days in the placebo group and 4 (3.0–9.5) days in the colchicine group (P = 0.604). The Kaplan-Meier curve revealed no difference in terms of clinical deterioration on day 14 (log-rank: P-value = 0.159) (Fig 2). Of the 9 patients who met the primary endpoint in the placebo group within 14 days, 5 required transfer to the ICU and

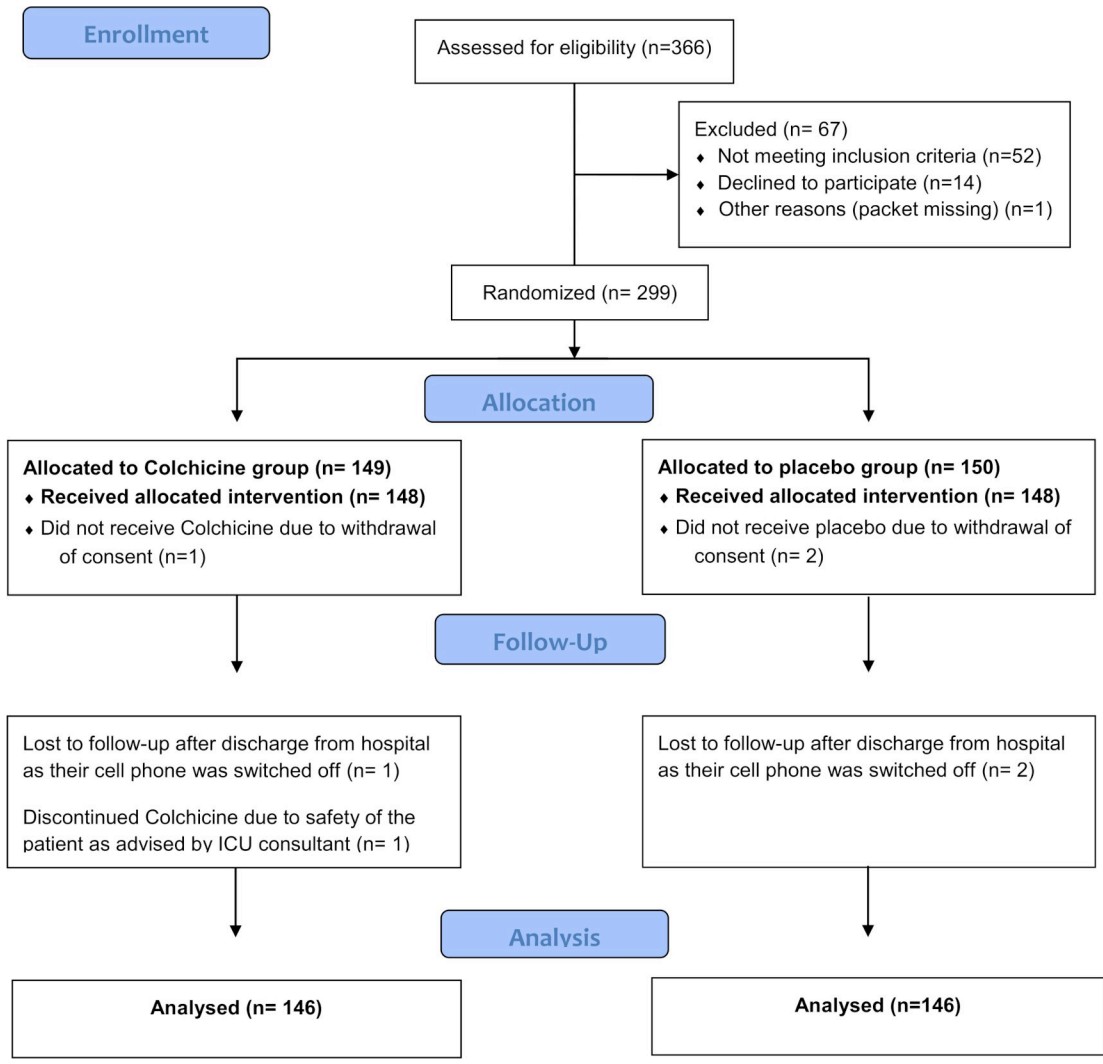

**Fig 1. Study flow diagram.**

mechanical support but died thereafter. The remaining four patients required respiratory support with HFNC at the HDU. A total of 4 participants in the colchicine group met the primary endpoints. Of them, 2 patients who died were transferred to the ICU and received mechanical ventilation. The other two patients needed respiratory support with HFNC in the HDU.

Among the secondary outcomes, the length of hospital stay after enrolment and the proportion of participants requiring supplemental oxygen were not found to significantly differ between the treatment groups. Colchicine might reduce the need mechanical ventilation [hazard ratio, (95% CI) - 0.49, (0.09 to 2.68)] and death [hazard ratio, (95% CI) - 0.39, (0.08 to 2.02)]; however, the values were not found to be statistically significant (Table 2).

The baseline blood biomarkers were compared with those found on days 7 and 14 of follow-up for patients in the colchicine and placebo groups. Both groups had statistically significant reductions in serum CRP levels. The changes in mean serum ferritin and D-dimer levels were not statistically significant (S1 Table).

**Table 1. Baseline characteristics of the study participants.**

| Characteristics | Total (n = 296) N (%) | Colchicine (n = 148) N (%) | Placebo (n = 148) N (%) |
|---|---|---|---|
| Age (Years) | | | |
| 18–40 | 109 (36.8) | 57 (38.5) | 52 (35.1) |
| 41–60 | 154 (52.0) | 75 (50.6) | 79 (53.4) |
| ≥61 | 33 (11.2) | 16 (10.8) | 17 (11.5) |
| Gender | | | |
| Female | 106 (35.8) | 53 (35.8) | 53 (35.8) |
| Male | 190 (64.2) | 95 (64.2) | 95 (64.2) |
| Duration (Day) of symptoms before enrolment Median (IQR) | 9 (7–12) | 9 (7–11) | 9 (6.25–12) |
| Clinical features | | | |
| Fever | 287 (97.0) | 143 (96.6) | 144 (97.3) |
| Cough | 177 (59.8) | 96 (64.9) | 81 (54.7) |
| Shortness of breath (SOB) | 134 (45.3) | 64 (43.2) | 70 (47.3) |
| Sore throat | 104 (35.1) | 48 (32.4) | 56 (37.8) |
| Diarrhea | 26 (8.8) | 13 (8.8) | 13 (8.8) |
| Anosmia | 1 (0.3) | 0 (0.0) | 1 (0.7) |
| Duration (Day) of symptoms before admission Median (IQR) | 6 (4–8) | 6 (4–8) | 6 (3–9) |
| Duration (Day) of fever at the time of Enrolment Median (IQR)$ | 9 (6–11) | 9 (7–11) | 8 (6–12) |
| Presence of Co-morbidity | 148 (50.0) | 69 (48.6) | 79 (51.3) |
| Diabetes mellitus | 100 (33.8) | 52 (35.1) | 48 (32.4) |
| Hypertension | 79 (26.7) | 32 (21.6) | 47 (31.8) |
| Chronic obstructive pulmonary disease (COPD)/Bronchial asthma | 23 (7.8) | 12 (8.1) | 11 (7.4) |
| Chronic kidney disease (CKD) | 3 (1.0) | 0 (0.0) | 3 (2.0) |
| Ischemic heart disease | 14 (4.8) | 4 (2.7) | 10 (6.8) |
| Chronic liver disease (CLD) | 2 (0.7) | 0 (0.0) | 2 (1.3) |
| Other Treatment | | | |
| Low molecular weight heparin (LMWH) | 260 (87.8) | 130 (87.8) | 130 (87.8) |
| Dexamethasone | 182 (61.5) | 97 (65.5) | 85 (57.4) |
| Ramdesivir | 27 (9.1) | 13 (8.7) | 14 (9.4) |
| Ceftriaxone | 73 (24.6) | 37 (25) | 36 (24.3) |
| Meropenem | 36 (12.2) | 16 (10.8) | 20 (13.5) |
| Baseline blood parameters Median (IQR) | | | |
| Hemoglobin* | 12.2(11.0–13.5) | 12.10(11.00–13.4) | 12.25(10.90–13.70) |
| Total count of WBC* | 8.5(6.2–11.8) | 8.5(6.4–12.2) | 8.4(6.1–11.1) |
| Neutrophil*(%) | 71.70(60.8–83.0) | 73.2(61.0–83.7) | 70.0(60.5–82.4) |
| Lymphocyte*(%) | 21.7(12.8–30.8) | 20.4(12.2–30.0) | 24.6(12.8–31.0) |
| Platelet count* | 247.0(188.5–324.0) | 245.0(191.5–328.5) | 249.0(185.2–315.0) |
| Serum Creatinine** | 1.0(0.8–1.1) | 0.93(0.80–1.10) | 0.99(0.81–1.12) |
| Random blood sugar (RBS)$ | 8.4(6.2–14.0) | 8.3(6.3–17.2) | 9(6.91–13.7) |
| Alanine aminotransferase (ALT)# | 40.0(24.0–59.0) | 49.0(29.0–71.0) | 34.5(22.0–60.0) |
| Serum C-reactive protein (CRP)@ | 12.0(6.0–36.3) | 10(6–28.20) | 15.24(6.00–25.02) |
| Serum Ferritin# | 297.0(121.5–672.0) | 301.0(175.0–758.0) | 256.0(95.04–642.0) |
| Lactate dehydrogenase (LDH)$ | 451.5(351.5–594.5) | 483.0(372.0–619.0) | 446.0(331.5–583.0) |

*(Continued)*

**Table 1.** (Continued)

| Characteristics | Total (n = 296)<br>N (%) | Colchicine (n = 148)<br>N (%) | Placebo (n = 148)<br>N (%) |
|---|---|---|---|
| D-dimer[&] | 0.5(0.3–1.1) | 0.43(0.26–0.83) | 0.51(0.28–1.50) |

$ median Duration (Days) of fever at the time of Enrolment for 288 records (144 in placebo and 144 in colchicine); ivermectin 1 patient in the placebo group.

* mean baseline levels for 294 records (148 in placebo and 148 in colchicine group); unit-gm/dl.

** mean baseline Serum creatinine for 284 records (140 in placebo and 144 colchicine).

$ mean baseline Random blood sugar (RBS) for 270 records (135 in placebo and 135 in colchicine); unit: mmol/Lt.

# mean baseline Alanine amino transferase (ALT) for 281 records (138 in placebo and 143 in colchicine).

@ mean baseline Serum Serum C-reactive protein (CRP) for 182 records (87 in placebo and 95 in colchicine).

# mean baseline Serum ferritin for 161 records (75 in placebo and 86 in colchicine).

$ mean baseline Lactate dehydrogenase (LDH) for 252 records (126 in placebo and 126 in colchicine).

& mean baseline d-Dimer for 191 records (99 in placebo and 92 in colchicine).

IQR = Inter quartile range.

## Subgroup analysis

Subgroup analysis of sex (male vs. female), age (18–40 years, 41–60 years, > 60 years), comorbidities (diabetes mellitus, hypertension, asthma, and COPD), and duration of symptoms before enrolment (10 days or less vs. more than 10 days) did not reveal any credible subgroup effects (S1 Fig).

## Additional analysis

The outcomes were analyzed on day 28 of follow-up. Two-point deterioration at the 28-day follow-up was found in 13 (8.9%) patients in the placebo group and 4 (2.7%) patients in the colchicine group. The rate of deterioration was significantly lower in colchicine group, with a hazard ratio [95%CI] of 0.29 [0.098–0.917], (P = 0.035). A Kaplan-Meier curve was constructed for this exploratory outcome on day 28, and the log-rank test revealed significance (P = 0.025) (Fig 3). This additional outcome is the same as all-cause mortality at day 28 because all patients who deteriorated by two or more points on the seven-category ordinal scale had a score of 7 (i.e., death). A total of 131 (89.7%) participants in the placebo group were at home on day 28 compared to 137 (93.8%) in the colchicine group (odds ratio [95%CI], 0.57 [0.24–1.35] P = 0.200).

## Safety outcome

We actively searched for three well-known side effects of colchicine, including diarrhea, nausea/vomiting, and abdominal pain. Further, we collected self-reported adverse events

**Table 2. Clinical outcomes of patients administered colchicine compared with those administered placebo on day 14 of follow-up.**

| Outcomes | Colchicine (N = 146)<br>n (%) | Placebo (N = 146)<br>n(%) | Hazard Ratio<br>(95% CI) | P-value |
|---|---|---|---|---|
| Decrease of 2 or more points in the ordinal outcome | 4 (2.7) | 9 (6.2) | 0.44 (0.13–1.43) | 0.171 |
| Participants requiring Supplemental oxygen (any device) | 7(4.8) | 7(4.8) | 0.98 (0.34–2.78) | 0.96 |
| Participants requiring mechanical ventilation (both non-invasive and invasive) | 2(1.4) | 4(2.7) | 0.49(0.09–2.68) | 0.41 |
| Death (all-cause mortality) | 2(1.4) | 5 (3.4) | 0.39(0.08–2.02) | 0.26 |
| Length of hospital stay Median (IQR) | 10 (7–15) | 9 (6–15) | | 0.59 |

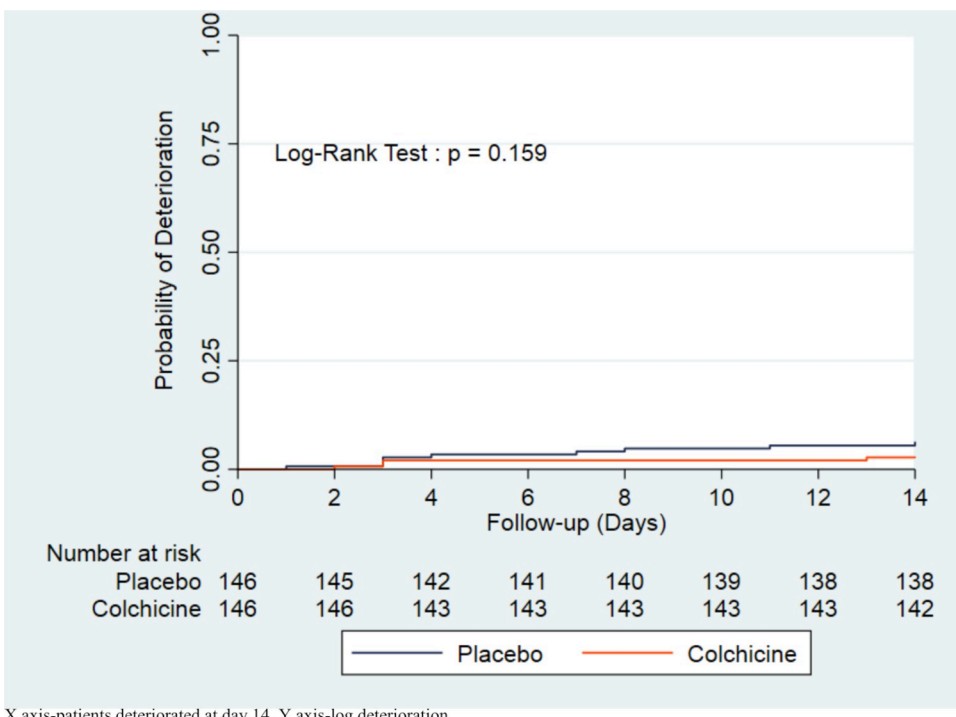

X axis-patients deteriorated at day 14, Y axis-log deterioration

**Fig 2. Kaplan-meier survival curves based on the primary clinical endpoint on day 14 using the log rank test.**

according to the Common Terminology Criteria for Adverse Events (CTCAE) version 5.0. Among the adverse events, diarrhea (18.5% in the colchicine group and 4.1% in the placebo group; P<0.001) and nausea/vomiting (8.2% in the colchicine group and 2.7% in the placebo group; P = 0.040) were significantly more frequent in the colchicine group than the control group (Table 3). However, these side effects were generally mild, self-limiting, and did not lead to drug discontinuation. None of the patients experienced dehydration, and diarrhea was controlled via oral ingestion of saline only. Omeprazole and domperidone were prescribed for nausea and vomiting. In addition, one patient in the colchicine group developed a macular skin rash on the trunk on day 3; however, this rash was ameliorated after 2 days with an oral antihistamine. At baseline, 20 (7.1%) patients had elevated serum ALT levels that exceeded three times the upper limit of normal: 11 (3.9%) in the colchicine group and 9 (3.2%) in the placebo group. The elevated serum ALT levels of 19 patients were close to normal at 14 days of follow-up. One patient (placebo group) with elevated serum ALT level at baseline showed a reduction at 10 days of follow-up but unfortunately died on day 11 in the intensive care unit due to acute respiratory distress syndrome. During treatment, patients with normal ALT levels at baseline did not show an elevation that would exceed three times the upper limit of normal.

Among the 4 patients who died in the colchicine group within the 28-day follow-up, 3 died at the hospital. The fourth patient clinically recovered and was discharged on day 14, but died suddenly on day 24. This patient was a 45 years old male with no chronic illness; however, his blood test revealed pancytopenia before enrolment. His pancytopenia did not improve during discharge from a specialized COVID-19 unit. As a result, this patient was referred to a hematology team. During the evaluation process, the patient died at home before a diagnosis was given. In the placebo group, 13 patients died within the 28 days of follow-up, 5 of whom died before day 14 of follow-up. Two patients who had a change in score of 5 on day 14 of follow-up died in the intensive care unit. Another patient who had a change in score of 5 on day 14 of

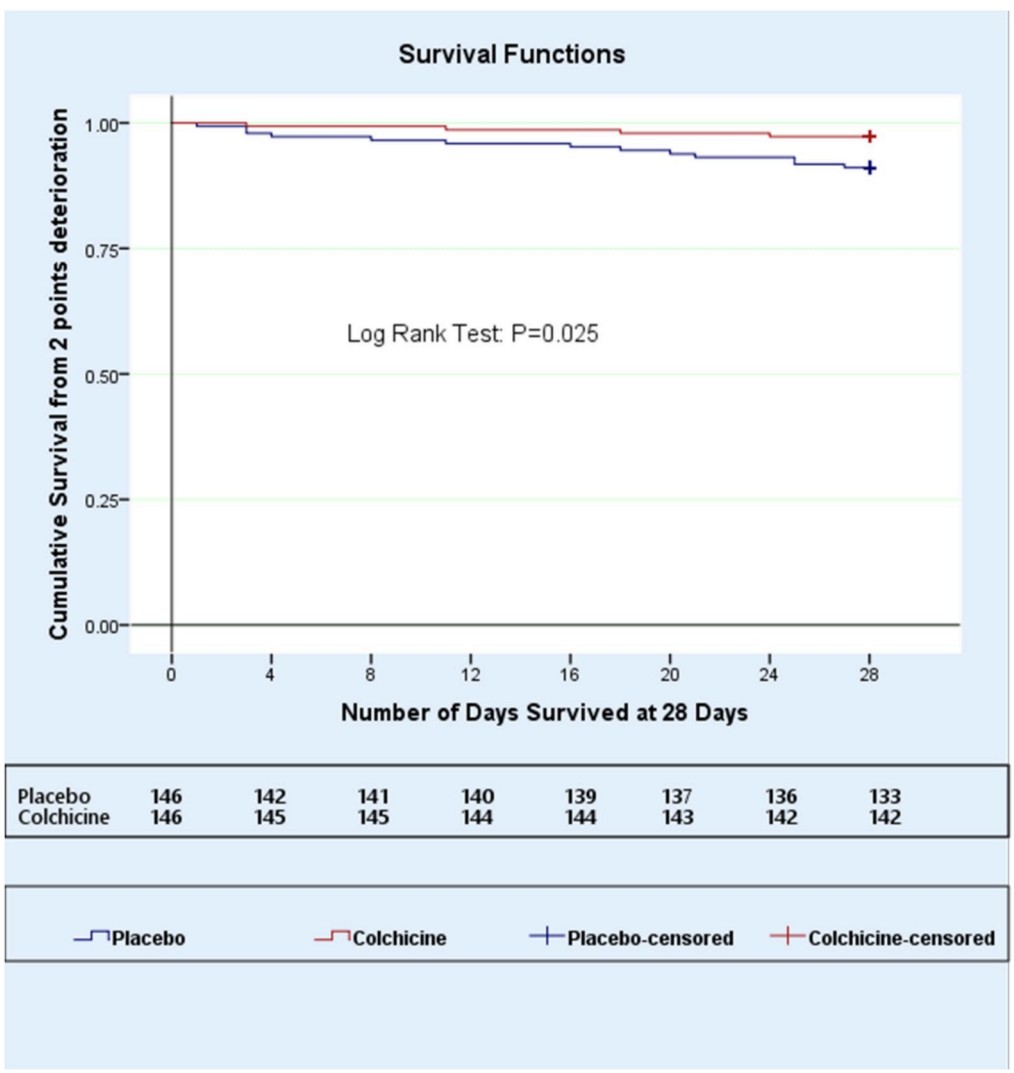

**Fig 3. Kaplan-meier survival curve on Day 28 of follow-up using the log-rank test.**

follow-up recovered clinically and was discharged from the hospital. Four patients whose clinical score was 3 or 4 during the day 14 follow-up deteriorated subsequently and died at the HDU or ICU; the other 2 patients showed clinical improvement and were discharged but died on days 20 and 25. One of these patients was a 37 years old female who was treated for breast cancer 2 years ago, and the other patient was a 60 years old male with diabetes mellitus.

## Discussion

Based on the findings of our study, there was a 56% reduction in the need for mechanical ventilation and death on day-14 in the colchicine arm, which was not statistically significant.

**Table 3. Comparison of adverse events between colchicine and placebo.**

| Adverse events | Colchicine (n = 146) | Placebo (n = 146) | P-value |
|---|---|---|---|
| Diarrhea, n(%) | 27 (18.5) | 6 (4.1) | <0.001 |
| Nausea/ vomiting, n(%) | 12 (8.2) | 4 (2.7) | 0.040 |
| Abdominal pain or burning, n(%) | 11 (7.5) | 6 (4.1) | 0.211 |

There was no significant difference in the length of hospital stay between groups. However, on day-28, colchicine significantly reduced the clinical deterioration by two or more points from baseline and all-cause mortality.

The open-level GRECCO trial suggested a significant clinical benefit of colchicine in hospitalized COVID-19 patients. The patient group in this trial had different severities of disease, ranging from categories 3 to 5, with a sample size of 105. The timeframe for the primary endpoint was 21 days [11]. In the COLCORONA trial, the outcome of patients with a positive RT-PCR test on day 30 following randomization revealed a significant reduction in the composite outcome of hospitalization and death [22]. The results of our study differed when the outcomes at day 14 were compared but were similar when the outcomes at day 28 of follow-up were compared. This finding indicates that the primary outcome may have been measured early, and the beneficial effect of colchicine during the cytokine storm may continue for the next two–three weeks.

Based on a recent meta-analysis of three RCTs, a benefit might exist in mortality that is not statistically significant among patients receiving colchicine versus non-colchicine regimens [23]. In our study, the same trends were found in the 14-day follow-up; however, in the 28 day follow-up, colchicine had a significant benefit.

In our study, the C-reactive protein (CRP) level was significantly reduced from the baseline on days 7 and 14 in both groups. Our result is similar to that of a previous study that revealed a significant reduction in serum CRP levels on days 4 and 7 compared to the baseline [13]. However, when we compared the median CRP level of the placebo group with that of the colchicine group on day 7, no significant difference was found. This was also true for the median CPR level on day 14. Our findings are consistent with those of the GRECCO trial [11]. However, it is not clear from this study whether the CRP decrease is related to the use of anti-inflammatory drugs, such as colchicine and dexamethasone, or just to the natural recovery of patients.

We enrolled hospitalized patients with moderate COVID-19 infection. In this patient group, colchicine did not have a beneficial effect on day 14 of treatment; however, a late beneficial effect was observed on day 28. Although this result can be generalized to this patient group in the population, this result may not be applicable to hospitalized patients with different severities and non-hospitalized patients.

## Limitations

This clinical trial had several limitations. This single-center study was conducted over a short period. Colchicine has a bitter taste that cannot be replicated in a placebo. Although patients were advised to swallow the tablets immediately after putting them into the mouth, the bitter taste of colchicine could have compromised the blinding to some extent. The patients were not hospitalized during the observation period. Accordingly, telecon follow-up was required, which has inherent drawbacks. Due to the limitations of the investigation facilities, in some instances, follow-up RT-PCR could not be performed for all participants at the scheduled time. Moreover, a follow-up investigation of the patient after discharge was not possible in most cases.

## Conclusion

Colchicine was not found to have a significant early beneficial effect on the reduction of mortality and the need for mechanical ventilation by hospitalized patients with moderate COVID-19; however, a late beneficial effect was observed. Further studies should be carried out to evaluate the late benefits.

## Supporting information

**S1 Checklist.**
(DOC)

**S1 Fig. Forest plot: Showing sub group effect at day 14.**
(DOCX)

**S1 Table. Comparison of baseline and follow up blood biomarkers between colchicine and placebo.**
(DOCX)

**S2 Table. Clinical outcome at day 14.**
(DOCX)

**S3 Table. Clinical outcome at day 28.**
(DOCX)

**S1 Protocol.**
(DOCX)

## Acknowledgments

The authors would like to acknowledge the contribution of Incepta Pharmaceuticals Ltd. 40, Shahid Tajuddin Ahmed Sarani, Tejgaon industrial area, Dhaka, Bangladesh, email: info@inceptapharma.com, for donating colchicine and placebo for our participants. We would also like to thank Editage (www.editage.com) for English language editing.

## Author Contributions

**Conceptualization:** Motlabur Rahman, Ponkaj K. Datta, Pratyay Hasan, Manjurul Haque, Khan Abul Kalam Azad, Titu Miah, Md. Mujibur Rahman.

**Data curation:** Khairul Islam, Mahfuzul Haque, Pratyay Hasan, Manjurul Haque, Imtiaz Faruq, Mohiuddin Sharif, Rifat H. Ratul.

**Formal analysis:** Motlabur Rahman, Ponkaj K. Datta, Reaz Mahmud, Pratyay Hasan.

**Funding acquisition:** Motlabur Rahman, Khan Abul Kalam Azad.

**Investigation:** Motlabur Rahman, Ponkaj K. Datta, Khairul Islam, Rifat H. Ratul.

**Methodology:** Motlabur Rahman, Ponkaj K. Datta, Khairul Islam, Mahfuzul Haque, Pratyay Hasan, Mohiuddin Sharif, Titu Miah, Md. Mujibur Rahman.

**Project administration:** Motlabur Rahman, Ponkaj K. Datta, Khan Abul Kalam Azad, Titu Miah, Md. Mujibur Rahman.

**Resources:** Motlabur Rahman, Ponkaj K. Datta, Uzzwal Mallik, Imtiaz Faruq, Mohiuddin Sharif, Khan Abul Kalam Azad, Md. Mujibur Rahman.

**Software:** Motlabur Rahman, Reaz Mahmud, Mohiuddin Sharif.

**Supervision:** Motlabur Rahman, Ponkaj K. Datta, Khairul Islam, Mahfuzul Haque, Uzzwal Mallik, Pratyay Hasan, Manjurul Haque, Imtiaz Faruq, Rifat H. Ratul, Khan Abul Kalam Azad, Md. Mujibur Rahman.

**Visualization:** Ponkaj K. Datta.

**Writing – original draft:** Motlabur Rahman, Ponkaj K. Datta.

**Writing – review & editing:** Motlabur Rahman, Ponkaj K. Datta, Khairul Islam, Mahfuzul
Haque, Reaz Mahmud, Uzzwal Mallik, Pratyay Hasan, Manjurul Haque, Imtiaz Faruq,
Rifat H. Ratul, Khan Abul Kalam Azad, Titu Miah, Md. Mujibur Rahman.

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
