## [Decision Letter · Decision Letter 0]

21 Dec 2021

PONE-D-21-32054Efficacy of Colchicine in moderate COVID-19 patients: A double-blind, randomized, placebo controlled trialPLOS ONE

Dear Dr. Datta,

Thank you for submitting your manuscript to PLOS ONE. After careful consideration, we feel that it has merit but does not fully meet PLOS ONE’s publication criteria as it currently stands. Therefore, we invite you to submit a revised version of the manuscript that addresses the points raised during the review process.

We look forward to receiving your revised manuscript.

Kind regards,

Gerald Chi, M.D.

Academic Editor

PLOS ONE

Journal Requirements: 

A clean copy of the edited manuscript (uploaded as the new *manuscript* file

Reviewers' comments:

Reviewer's Responses to Questions

**Comments to the Author**

1. Is the manuscript technically sound, and do the data support the conclusions?

Reviewer #1: Partly

Reviewer #2: Yes

Reviewer #3: Partly

2. Has the statistical analysis been performed appropriately and rigorously? 

Reviewer #1: N/A

Reviewer #2: Yes

Reviewer #3: Yes

3. Have the authors made all data underlying the findings in their manuscript fully available?

Reviewer #1: Yes

Reviewer #2: Yes

Reviewer #3: Yes

4. Is the manuscript presented in an intelligible fashion and written in standard English?

Reviewer #1: Yes

Reviewer #2: Yes

Reviewer #3: No

5. Review Comments to the Author

Reviewer #1: This is an interesting RCT done in Bangladesh to evaluate the efficacy of colchicine in reducing Covid-19 severity in hospitalized patients. So far, the summary of the published evidence points to colchicine having no influence on mortality or progression in people hospitalized with moderate or severe Covid, and colchicine having an uncertain effect on people with mild or asymptomatic diseases. Therefore, more evidence is welcome.

My main concern for this study is that the severity of Covid-19 patients on enrolment is not very clear, as despite including only patients requiring hospitalization, less than 5% of patients needed oxygen in both arms during hospital stay. So this is quite unusual unless they were all hospitalized based on other-medical reasons, these participants behave like an outpatient setting. Furthermore, about 60% of patients received dexamethasone, which would have been incorrectly prescribed if over 95% of patients did not require oxygen. Also, the sample size calculation does not seem appropriate as further detailed below.

INTRODUCTION.

• The authors state that there is no effective treatment for Covid-19 inflammatory response; however, corticosteroids have been shown to reduce mortality in COVID pneumonia (Recovery trial)

• Also, there are several systematic reviews on Covid-19 and colchicine. The most comprehensive and rigorous is a recent COCHRANE review (doi: 10.1002/14651858.CD015045) evaluating colchicine for the treatment of Covid-19. It included three RCTs with over 11 thousand hospitalized participants and concluded that based on the current evidence, in people hospitalized with moderate to severe COVID-19 the use of colchicine probably has little to no influence on mortality or clinical progression. However, there is uncertain evidence of the effect of colchicine on mortality for people with asymptomatic infection or mild disease and colchicine probably results in a slight reduction of hospital admissions or deaths within 28 days, and the rate of serious adverse events compared with placebo. These results from the systematic review should be included in the manuscript.

METHODS

• Page 16 of Study Protocol in supplementary information states, "All patients will be in category 3, hospitalized, not requiring supplemental oxygen; according to seven-category ordinal scale. Patients with other categories will not be included in this study”. This should also be mentioned equally clearly in the main paper.

• It is not clear if radiologic pneumonia was a required criterion for enrolment or not. This is not mentioned in the paper, although it was included as an inclusion criterion in clinical.trials registry.

• Concerns regarding sample size calculation:

o The primary outcome was “time to develop clinical deterioration”, defined as the time from randomization to a deterioration of two or more points (from the status at randomization) on a seven-category ordinal scale. However, sample size calculation was done based on the "proportion of subjects that would be cured by colchicine”. Therefore, sample size calculation does not seem to have been calculated on the same outcome than the primary outcome.

o Additionally, the authors provide no reference for the “assumption that the proportion of subjects that would be cured by colchicine is 50%” nor for the estimated “16%” difference for the outcome between both study arms.

RESULTS:

• The proportion of pneumonia in chest images in each group is not shown, although all participants had a chest image done per protocol.

• Remdesevir use was allowed per-protocol but is not reported in table 1

• There is an error in the calculation for the proportion of participants having received dexamethasone in both study arms and reported in Table 1: colchicine arm 97/148 = 65.5% (and not 32.8%) and in control arm 85/148 =57.4% (and not 28.7%). The same miscalculation is found for the LMWH proportion of use in both arms in Table 1.

• It is striking that in a clinical trial of patients hospitalized for Covid-19, less than 5% (14 of 292 as shown in Table 2) of all participants required supplemental oxygen, and less than 4.5 % deteriorated (primary outcome), considering that mean day of hospital stay was 11 days.

Also, if sample size calculation assumption was that 50% would deteriorate in control arm, this assumption was very far from what really happened and the estimation was not met.

• Three deaths occurred after hospital discharge (1 in colchicine arm, two in control arm) and no information is provided.

• It is unclear why the authors did not do a multivariate analysis.

• English writing could be improved.

Reviewer #2: Efficacy of Colchicine in moderate COVID-19 patients: A double-blind, randomized, placebo controlled trial

PONE-D-21-32054

General comment

An important piece of work which is investigator-initiated and commendable. I have a couple of major and minor points for the consideration of the authors

Major comments

• Role of Incepta Pharmaceuticals Ltd

The authors make the point that the colchicine and the placebo was supplied by Incepta Pharmaceuticals Ltd. 40, Shahid Tajuddin Ahmed Srani, Tejgaon industrial area, Dhaka, Bangladesh …….. and the company had no role in planning, design, data collection, analysis …….

This turns out not to be exactly consistent with further comments. The authors report that the random numbers were generated and maintained by an independent biostatistician of Incepta pharmaceuticals Limited.

It the above is true, that represents substantial involvement in the drug producer in the design and implementation of the trial, and a case for conflict of interest.

• How the arms were treated

It is important that how patients in the arms of the study were treated be thoroughly detailed. Differences in care (beside the intervention) could account for the observed outcomes. Accordingly, I suggest that the distribution between the arms in terms of treatment as below is fully detailed in a table.

Additionally, Paracetamol, antihistamines, oxygen therapy were also given as part of standard care according to the National guideline of Bangladesh and Clinical Management of COVID-19, Interim Guidance of World Health Organization [13][14]. Low molecular weight heparin, according to the indication, appropriate broad-spectrum antibiotics-if needed, Remdesivir injection as well as other drugs for associated comorbid conditions were also prescribed by the treating physicians.

• Blinding

Colchicine has a very bitter taste. How did the investigator address this in the production of the placebos to preserve blinding?

• Inclusion criteria and endpoint

The study enrolled patients with oxygen concentration of no less than 94% without supplemental oxygen. How then did oxygen saturation greater than 93% without supplemental oxygen become an endpoint? That endpoint was achieved even at the time of enrolment.

• Presentation

All abbreviations in the tables should be written in full or have a key to guide readers.

• Primary and secondary outcomes

The sample size statement should make clear whether the assumptions are based on outcomes at Day 14 or 28.

The statement “Colchicine significantly reduced the primary outcome and all-cause mortality” is inaccurate as day 28 analysis was outcome of secondary analysis as per statements in the abstract & in line 242.

Minor comments

• State terms in full when they are first used e.g. Colchicine inhibits IL-1β, NLRP3 activation, ARDS, JAK ….…

• Colchicine need not be come with a capital C.

• Pneumonia need not have a capital P in “World Health Organization (WHO) made notification of the Pneumonia case-cluster”

• Cytokine need not have a capital C in “The Cytokine storm syndrome….”

• It is unclear what benefit the untitled table in lines 183-4 add to the write-up. It can be well described in the text.

• It is possible to make the abstract less wordy e.g. “Data were collected using a case-record form.” Can be removed.

Reviewer #3: A randomized clinical trial was conducted which aimed to determine whether adding colchicine with other standard of care treatment improved moderate COVID-19 pneumonia. A late beneficial effect of colchicine was observed compared to standard of care.

Minor revisions:

1- Line 250: Is “cured” the correct terminology?

2- Line 248: Indicate the statistical testing method which achieves 80% power.

3- Line 257: Perhaps the more precise terminology is “dependent” instead of “related.”

4- Line 273: Remove the comma between December and 15.

5- Table 1: Clarify the results reported for the placebo group’s duration of symptoms.

6- Line 325: Replace “across” with “between” since there are only two groups.

7- Indicate if adverse events were collected according to a standardized method, such as CTCAE.

8- Figure 2: To more clearly display the time to event data, cut the y-axis at 0.25.

9- Figure 3: (a) Provide a more descriptive y-axis label. (b) What days do the number at risk correspond to? The layout makes this difficult to determine.

10- Thoroughly proofread the manuscript.

6. PLOS authors have the option to publish the peer review history of their article (what does this mean?). If published, this will include your full peer review and any attached files.

Reviewer #1: No

Reviewer #2: **Yes: **Frank Baiden

Reviewer #3: No

---

## [Author Response · Author response to Decision Letter 0]

2 Apr 2022

To 

Editor in chief,

PLOS ONE

Subject: In response to review of the manuscript entitled “Efficacy of colchicine in moderate COVID-19 patients: A double-blind, randomized, placebo controlled trial”.

Dear Sir,

Thank you for reviewing my manuscript. I have tried to address each point raised by the academic editor and the reviewers.

Response to Academic editor

1. PLOS ONE style-I have revised the manuscript according to PLOS ONE style.

2. The manuscript was edited by Editage. A copy was uploaded as supporting information. A clean copy of edited manuscript was uploaded as manuscript file.

Review Comments to the Author 

Reviewer #1: This is an interesting RCT done in Bangladesh to evaluate the efficacy of colchicine in reducing Covid-19 severity in hospitalized patients. So far, the summary of the published evidence points to colchicine having no influence on mortality or progression in people hospitalized with moderate or severe Covid, and colchicine having an uncertain effect on people with mild or asymptomatic diseases. Therefore, more evidence is welcome.

My main concern for this study is that the severity of Covid-19 patients on enrolment is not very clear, as despite including only patients requiring hospitalization, less than 5% of patients needed oxygen in both arms during hospital stay. So this is quite unusual unless they were all hospitalized based on other-medical reasons, these participants behave like an outpatient setting. Furthermore, about 60% of patients received dexamethasone, which would have been incorrectly prescribed if over 95% of patients did not require oxygen. Also, the sample size calculation does not seem appropriate as further detailed below.

Author’s response: Thank you. It is true that these participants behave like an out patients setting. We enquired into the causes of their hospitalization and found the following reasons- 

1) Among the participants, 148 (50%) patients had at least one of the comorbid conditions like diabetes mellitus, hypertension, COPD/ bronchial asthma, chronic kidney disease, chronic liver disease, ischemic heart disease. 

2) Along with the moderate COVID-19 disease some patients also admitted either for the management of the coexisting disease like rheumatoid arthritis, SLE, epilepsy, myelo-proliferative disease or for the evaluation of co-existing clinical feature like hepatomegaly, central chest pain, lower limb weakness, abdominal pain (later diagnosed as cholecystitis), pancytopenia. 

3) More than 11% of our study participants are from the age group of 61 years or more. Considering their risk of development of the severe disease these patients were admitted during the initial phase of the pandemic. 

4) Some patients with moderate COVID-19 were also hospitalized due to the severity of their symptoms like intense and irritating cough, high fever for the duration of 7 or more days, fever not responding to paracetamol. 8.8% of the patients presented with diarrhea with variable degree of dehydration which leads to their hospitalization. 

5) Less than 5% of our participants are health care workers, despite having moderate COVID-19 they were hospitalized as part of providing institutionalized isolation facilities. 

Concern regarding less than 5% of patients needed oxygen- during 14th day follow up we found only 14 (on oxygen) plus 7 (death-also required oxygen) patients i,e 21 patients (7.2%) required oxygen. When we analyzed the highest in hospital severity in terms of 7 category ordinal scale, we found that 68 (23%) patients required supplemental oxygen at least for few hours during their hospital stay. 

Concern about 60% of patients received dexamethasone- about 23% of the patients needed supplemental oxygen in our study but about 60% received dexamethasone. We appreciate your concern and agree with you that the use of dexamethasone might be inappropriate during the initial period of the trial. We started patient enrolment from 15th July, 2020 and WHO provide the guidance regarding dexamethasone use during the first week of September, 2020. During the initial phase of the pandemic there was a tendency to use dexamethasone based on few preprint reports with the intention to help patients with whatever evidence we have.

INTRODUCTION.

• The authors state that there is no effective treatment for Covid-19 inflammatory response; however, corticosteroids have been shown to reduce mortality in COVID pneumonia (Recovery trial)

Author’s response: thank you. Information is added in the manuscript. (Line-112)

Also, there are several systematic reviews on Covid-19 and colchicine. The most comprehensive and rigorous is a recent COCHRANE review (doi: 10.1002/14651858.CD015045) evaluating colchicine for the treatment of Covid-19. It included three RCTs with over 11 thousand hospitalized participants and concluded that based on the current evidence, in people hospitalized with moderate to severe COVID-19 the use of colchicine probably has little to no influence on mortality or clinical progression. However, there is uncertain evidence of the effect of colchicine on mortality for people with asymptomatic infection or mild disease and colchicine probably results in a slight reduction of hospital admissions or deaths within 28 days, and the rate of serious adverse events compared with placebo. These results from the systematic review should be included in the manuscript.

Author’s response: Thank you. Information from COCHRANE review is added in the manuscript. (Line-123-132)

METHODS

• Page 16 of Study Protocol in supplementary information states, "All patients will be in category 3, hospitalized, not requiring supplemental oxygen; according to seven-category ordinal scale. Patients with other categories will not be included in this study”. This should also be mentioned equally clearly in the main paper.

Author’s response: Thank you. This is added under ‘patients and methods’ heading from line 180 to 182. 

• It is not clear if radiologic pneumonia was a required criterion for enrolment or not. This is not mentioned in the paper, although it was included as an inclusion criterion in clinical.trials registry.

Author’s response: thank you. This is added under ‘patients’ heading from line 173 to 174. 

Concerns regarding sample size calculation:

 The primary outcome was “time to develop clinical deterioration”, defined as the time from randomization to a deterioration of two or more points (from the status at randomization) on a seven-category ordinal scale. However, sample size calculation was done based on the "proportion of subjects that would be cured by colchicine”. Therefore, sample size calculation does not seem to have been calculated on the same outcome than the primary outcome.

o Additionally, the authors provide no reference for the “assumption that the proportion of subjects that would be cured by colchicine is 50%” nor for the estimated “16%” difference for the outcome between both study arms.

Author’s response: Thank you. Necessary correction has been made in the respective section. (Line-275-291)

RESULTS:

The proportion of pneumonia in chest images in each group is not shown, although all participants had a chest image done per protocol.

Author’s response: Thank you. Proportion of pneumonia in chest images was not recorded in the case record form. We enrolled the patients if CT chest involvement is less than 50%. In case of chest x-ray, patients were enrolled only if at least two of the investigators agreed that lung field involvement is less than 50%, as percentage of involvement in chest x-ray is not routinely reported by radiologist in our settings.

Remdesevir use was allowed per-protocol but is not reported in table 1

Author’s response: Thank you. This is added in table 1. 

• There is an error in the calculation for the proportion of participants having received dexamethasone in both study arms and reported in Table 1: colchicine arm 97/148 = 65.5% (and not 32.8%) and in control arm 85/148 =57.4% (and not 28.7%). The same miscalculation is found for the LMWH proportion of use in both arms in Table 1.

Author’s response: Thank you. This is corrected in table 1. 

 It is striking that in a clinical trial of patients hospitalized for Covid-19, less than 5% (14 of 292 as shown in Table 2) of all participants required supplemental oxygen, and less than 4.5 % deteriorated (primary outcome), considering that mean day of hospital stay was 11 days.

Also, if sample size calculation assumption was that 50% would deteriorate in control arm, this assumption was very far from what really happened and the estimation was not met.

Author’s response: Thank you. During 14th day follow up we found only 14 (on oxygen) plus 7 (death-also required oxygen) patients i,e 21 patients (7.2%) required oxygen. When we analyzed the highest in hospital severity in terms of 7 category ordinal scale, we found that 68 (23%) patients required supplemental oxygen at least for few hours during their hospital stay. 

Assumption about 50% deterioration to scale 5 or more in the 7-category ordinal scale in control arm was made considering the presence of comorbidities and the age of the patients which are responsible for the hospitalization of the moderate COVID-19 patients. Also, the course and the prognosis were not established like now a day. However, we agree with you that the assumption was far from the reality. 

Three deaths occurred after hospital discharge (1 in colchicine arm, two in control arm) and no information is provided.

Author’s response: Thank you. As they died at home their cause of death could not be established but information is added from line 424 to 435. 

• It is unclear why the authors did not do a multivariate analysis.

Author’s response: thank you. Multivariate analysis was done but the authors did not include them in the manuscript as they thought it would add very little new information. 

• English writing could be improved.

Author’s response: Thank you. We have edited our writing. 

Reviewer #2: Efficacy of Colchicine in moderate COVID-19 patients: A double-blind, randomized, placebo-controlled trial

PONE-D-21-32054

General comment

An important piece of work which is investigator-initiated and commendable. I have a couple of major and minor points for the consideration of the authors

Major comments

• Role of Incepta Pharmaceuticals Ltd

The authors make the point that the colchicine and the placebo was supplied by Incepta Pharmaceuticals Ltd. 40, Shahid Tajuddin Ahmed Srani, Tejgaon industrial area, Dhaka, Bangladesh …….. and the company had no role in planning, design, data collection, analysis …….

This turns out not to be exactly consistent with further comments. The authors report that the random numbers were generated and maintained by an independent biostatistician of Incepta pharmaceuticals Limited.

It the above is true, that represents substantial involvement in the drug producer in the design and implementation of the trial, and a case for conflict of interest.

Author’s response: thank you. Necessary corrections made in line 162-164 and ‘conflict of interest’ section. 

• How the arms were treated

It is important that how patients in the arms of the study were treated be thoroughly detailed. Differences in care (beside the intervention) could account for the observed outcomes. Accordingly, I suggest that the distribution between the arms in terms of treatment as below is fully detailed in a table.

Additionally, Paracetamol, antihistamines, oxygen therapy were also given as part of standard care according to the National guideline of Bangladesh and Clinical Management of COVID-19, Interim Guidance of World Health Organization [13][14]. Low molecular weight heparin, according to the indication, appropriate broad-spectrum antibiotics-if needed, Remdesivir injection as well as other drugs for associated comorbid conditions were also prescribed by the treating physicians.

Author’s response: thank you. Distribution of Remdesivir, injectable antibiotics like ceftriaxone, meropenem is added in table 1. 

• Blinding

Colchicine has a very bitter taste. How did the investigator address this in the production of the placebos to preserve blinding?

Author’s response: thank you. Only excipients were used to produce the placebo. Nothing was added to make the placebo bitter. We advised the patients to take the tablet with water and to ingest before melting in mouth. 

• Inclusion criteria and endpoint 

The study enrolled patients with oxygen concentration of no less than 94% without supplemental oxygen. How then did oxygen saturation greater than 93% without supplemental oxygen become an endpoint? That endpoint was achieved even at the time of enrolment.

Author’s response: thank you. ‘Oxygen saturation greater than 93% without supplemental oxygen’ was not an endpoint. 

• Presentation

All abbreviations in the tables should be written in full or have a key to guide readers.

Author’s response: thank you. Abbreviations are written in full below the table. 

• Primary and secondary outcomes

The sample size statement should make clear whether the assumptions are based on outcomes at Day 14 or 28.

The statement “Colchicine significantly reduced the primary outcome and all-cause mortality” is inaccurate as day 28 analysis was outcome of secondary analysis as per statements in the abstract & in line 242.

Author’s response: thank you. Necessary correction has been made in the manuscript. 

Minor comments

• State terms in full when they are first used e.g. Colchicine inhibits IL-1β, NLRP3 activation, ARDS, JAK ….…

Author’s response: thank you. Terms are stated in full at their first appearance. 

• Colchicine need not be come with a capital C.

• Pneumonia need not have a capital P in “World Health Organization (WHO) made notification of the Pneumonia case-cluster”

• Cytokine need not have a capital C in “The Cytokine storm syndrome….”

• It is unclear what benefit the untitled table in lines 183-4 add to the write-up. It can be well described in the text.

• It is possible to make the abstract less wordy e.g. “Data were collected using a case-record form.” Can be removed.

Author’s response: thank you. All suggested correction has been made in the manuscript. 

Reviewer #3: A randomized clinical trial was conducted which aimed to determine whether adding colchicine with other standard of care treatment improved moderate COVID-19 pneumonia. A late beneficial effect of colchicine was observed compared to standard of care.

Minor revisions:

1- Line 250: Is “cured” the correct terminology?

2- Line 248: Indicate the statistical testing method which achieves 80% power.

3- Line 257: Perhaps the more precise terminology is “dependent” instead of “related.”

4- Line 273: Remove the comma between December and 15.

5- Table 1: Clarify the results reported for the placebo group’s duration of symptoms.

6- Line 325: Replace “across” with “between” since there are only two groups.

7- Indicate if adverse events were collected according to a standardized method, such as CTCAE.

8- Figure 2: To more clearly display the time to event data, cut the y-axis at 0.25.

9- Figure 3: (a) Provide a more descriptive y-axis label. (b) What days do the number at risk correspond to? The layout makes this difficult to determine.

10- Thoroughly proofread the manuscript.

Author’s response to minor revisions: thank you for nice suggestions. Corrections have been made accordingly. 

I hope I have tried my level best to address all of your point raised during the review process. Please consider my manuscript for publication in PLOS ONE.

Thanks

Dr. Ponkaj Kanti Datta

Assistant professor

Department of medicine

Dhaka Medical College

---

## [Decision Letter · Decision Letter 1]

10 May 2022

PONE-D-21-32054R1Efficacy of colchicine in patients with moderate COVID-19: A double-blinded, randomized, placebo-controlled trialPLOS ONE

Dear Dr. Datta,

Thank you for submitting your manuscript to PLOS ONE. After careful consideration, we feel that it has merit but does not fully meet PLOS ONE’s publication criteria as it currently stands. Therefore, we invite you to submit a revised version of the manuscript that addresses the points raised during the review process.

We look forward to receiving your revised manuscript.

Kind regards,

Miquel Vall-llosera Camps

Senior Editor

PLOS ONE

Additional Editor Comments:

The reviewers are mostly positive about your revised manuscript, but have raised remaining concerns that need to be addressed in a revision, in particular please consider Reviewer#1 comments.

Reviewers' comments:

Reviewer's Responses to Questions

**Comments to the Author**

1. If the authors have adequately addressed your comments raised in a previous round of review and you feel that this manuscript is now acceptable for publication, you may indicate that here to bypass the “Comments to the Author” section, enter your conflict of interest statement in the “Confidential to Editor” section, and submit your "Accept" recommendation.

Reviewer #1: (No Response)

Reviewer #2: All comments have been addressed

Reviewer #3: (No Response)

2. Is the manuscript technically sound, and do the data support the conclusions?

Reviewer #1: Partly

Reviewer #2: Yes

Reviewer #3: Yes

3. Has the statistical analysis been performed appropriately and rigorously? 

Reviewer #1: Yes

Reviewer #2: Yes

Reviewer #3: Yes

4. Have the authors made all data underlying the findings in their manuscript fully available?

Reviewer #1: Yes

Reviewer #2: Yes

Reviewer #3: Yes

5. Is the manuscript presented in an intelligible fashion and written in standard English?

Reviewer #1: Yes

Reviewer #2: Yes

Reviewer #3: Yes

6. Review Comments to the Author

Reviewer #1: Major comments:

1. Outcomes at 28 days were not included in Primary nor in Secondary study outcomes as defined in Methods section line 224-225 (“The time frame for all outcomes was 14 days post-randomization”). Therefore additional outcomes at 28 days should be considered exploratory and not “primary outcomes”. Despite this issue was already raised by Reviewer nº2, the manuscripts state in several paragraphs that on day 28, colchicine “significantly reduced the primary outcome” and all-cause mortality. This must be carefully corrected in the Abstract (lines 65, 68), Results section (lines 332, 334, 336) and in the Discussion (line 381) section.

2. I have commented in my previous revision this: “The proportion of pneumonia in chest images in each group is not shown, although all participants had a chest image done per protocol.”

The authors responded: “Proportion of pneumonia in chest images was not recorded in the case record form. We enrolled the patients if CT chest involvement is less than 50%. In case of chest x-ray, patients were enrolled only if at least two of the investigators agreed that lung field involvement is less than 50%, as percentage of involvement in chest x-ray is not routinely reported by radiologist in our settings.”

I am sorry that my question wasn't clear enough. What I wanted to say is that it is important that the authors show how many participants in each study arm did have radiologic pneumonia on admission (the proportion of participants having pneumonia in each group), not the area or amount of radiologic involvement. This information is important for the readers, and if available, it should be included in Table 1.

Minor comments:

3. Line 93, add “inhibitors” after IL-1

4. Line 113: remove the extra period after COVID-19

5. Line 153-155: “All patients enrolled in this study were assigned to category 3, were hospitalized, and did not require supplemental oxygen, according to a seven-category ordinal scale.” The readers do not know at that point of the manuscript to which category the authors are referring until much later in the manuscript. I suggest to add a reference to WHO categories or refering in brackets something as “as defined in the Outcomes Measures section”.

6. Line 169 (last line of the Blinding Table) states that “statistician” were not blind. Were these statisticians also involved in data results analysis? What was the role of this “unblind” statisticians? This is confusing

7. The authors explained in their response to reviewers that the placebo had the same excipients than colchicine pills, but in Line 184 the authors state that “placebo tasted similar to the study drugs”. Having the same excipients does not necessarily give the same bitter taste.

8. Line 264, add the year 2020 after December 15.

9. Table 1 shows results for 296 participants but 299 were randomized, this difference should be explained in the results section, line 265

10. Table 1, under comorbidities, several abbreviations, such as CLD, CKD, COPD are undefined. This was already mentioned by Reviewer nº2.

11. Was body mass index, or obesity measured in both groups? This is an important risk factor for severe COVID-19

12. Line 332, replace P-0.0035 with p=0.0035

13. Line 331, remove upper-case from “Colchicine” and “Hazard” ratio

14. Line 335 “ALT” is not previously defined

15. Line 355-358 the description regarding liver testing abnormalities in both study arms is a little confusing, I suggest revising the writing

16. Discussion, line 379-380. The authors state that “colchicine-treated patients deteriorated 1 day later than the placebo group”. This finding was not statistically significant, therefore it should be removed.

17. Discussion: Line 383-389, revise text wording, several sentences are not well written

18. In general, the Discussion could be revised and much improved. For example, in line 396-402 all the discussion about CRP levels is unclear, I suggest revising carefully. How can the authors conclude that the CRP decrease relates to anti-inflammatory drug use and not just on clinical natural recovery of patients? This looks speculative.

19. Any further discussion on the limitation of having over 2/3 of participants on both arms having received dexamethasone? Do the authors think that colchicine may have an additive beneficial effect ? (this is where a multivariate analysis may have helped in the results interpretation…)

Reviewer #2: The authors have done well to address the comments made by reviewers. I believe the manuscript is a a stage where readers will be able to judge well its merits and demerits. Regarding blinding, I will recommend to the authors to consider admitting that the bitter taste of colchicine could have compromised the effort to blind.

Reviewer #3: Minor Revision:

Figure 3: What time points do the number at risk correspond to? The layout makes this difficult to determine. Placing the number at risk immediately below the Number of Days Survived on the x-axis may clarify.

7. PLOS authors have the option to publish the peer review history of their article (what does this mean?). If published, this will include your full peer review and any attached files.

Reviewer #1: No

Reviewer #2: No

Reviewer #3: No

---

## [Author Response · Author response to Decision Letter 1]

13 Jul 2022

To 

Editor in chief,

PLOS ONE

Subject: In response to review of the manuscript entitled “Efficacy of colchicine in moderate COVID-19 patients: A double-blind, randomized, placebo controlled trial”.

Dear Sir,

Thank you for reviewing my manuscript. I have tried to address each point raised by the academic editor and the reviewers.

Response to Academic editor:

1. PLOS ONE style-I have revised the manuscript according to PLOS ONE style.

2. The manuscript was edited by Editage. A copy was uploaded as supporting information. A clean copy of edited manuscript was uploaded as manuscript file.

Additional Editor Comments:

The reviewers are mostly positive about your revised manuscript, but have raised remaining concerns that need to be addressed in a revision, in particular please consider Reviewer#1 comments.

Response to the editor:

Thank you very much. We have tried to address the concerns raised by the respected reviewer’s. 

Reviewers' comments:

Reviewer's Responses to Questions

Comments to the Author

1. If the authors have adequately addressed your comments raised in a previous round of review and you feel that this manuscript is now acceptable for publication, you may indicate that here to bypass the “Comments to the Author” section, enter your conflict of interest statement in the “Confidential to Editor” section, and submit your "Accept" recommendation.

Reviewer #1: (No Response)

Reviewer #2: All comments have been addressed

Reviewer #3: (No Response)

2. Is the manuscript technically sound, and do the data support the conclusions?

Reviewer #1: Partly

Reviewer #2: Yes

Reviewer #3: Yes

3. Has the statistical analysis been performed appropriately and rigorously? 

Reviewer #1: Yes

Reviewer #2: Yes

Reviewer #3: Yes

4. Have the authors made all data underlying the findings in their manuscript fully available?

Reviewer #1: Yes

Reviewer #2: Yes

Reviewer #3: Yes

5. Is the manuscript presented in an intelligible fashion and written in standard English?

Reviewer #1: Yes

Reviewer #2: Yes

Reviewer #3: Yes

6. Review Comments to the Author

Reviewer #1: Major comments:

1. Outcomes at 28 days were not included in Primary nor in Secondary study outcomes as defined in Methods section line 224-225 (“The time frame for all outcomes was 14 days post-randomization”). Therefore additional outcomes at 28 days should be considered exploratory and not “primary outcomes”. Despite this issue was already raised by Reviewer nº2, the manuscripts state in several paragraphs that on day 28, colchicine “significantly reduced the primary outcome” and all-cause mortality. This must be carefully corrected in the Abstract (lines 65, 68), Results section (lines 332, 334, 336) and in the Discussion (line 381) section.

Author’s response: thank you very much. The author’s are agreed with you. We are sorry for missing to respond properly to the previous reviewer. We have made necessary corrections in the above mentioned lines. 

2. I have commented in my previous revision this: “The proportion of pneumonia in chest images in each group is not shown, although all participants had a chest image done per protocol.”

The authors responded: “Proportion of pneumonia in chest images was not recorded in the case record form. We enrolled the patients if CT chest involvement is less than 50%. In case of chest x-ray, patients were enrolled only if at least two of the investigators agreed that lung field involvement is less than 50%, as percentage of involvement in chest x-ray is not routinely reported by radiologist in our settings.”

I am sorry that my question wasn't clear enough. What I wanted to say is that it is important that the authors show how many participants in each study arm did have radiologic pneumonia on admission (the proportion of participants having pneumonia in each group), not the area or amount of radiologic involvement. This information is important for the readers, and if available, it should be included in Table 1.

Author’s response: thank you. According to inclusion criteria we included the patients with moderate symptomatic pneumonia. According to National Guidelines on Clinical Management of Coronavirus Disease 2019 (COVID- 19),Version 7.0 .2020, directorate general of health services (DGHS), ministry of health and family welfare (MOHFW), Government of the People's Republic of Bangladesh, moderate pneumonia patients must have radiologic pneumonia but the involvement would be less than 50% (refer to protocol –page 24). So, all the enrolled patients of this study had radiologic pneumonia. We are sorry that, it was not clear in our previous manuscript. We added “(i.e. all the patients of both arms had radiologic pneumonia during enrolment)” at line 149 in “patients” section for further clarification. 

Minor comments:

3. Line 93, add “inhibitors” after IL-1

Author’s response: thank you. Added

4. Line 113: remove the extra period after COVID-19

Author’s response: thank you. Removed

5. Line 153-155: “All patients enrolled in this study were assigned to category 3, were hospitalized, and did not require supplemental oxygen, according to a seven-category ordinal scale.” The readers do not know at that point of the manuscript to which category the authors are referring until much later in the manuscript. I suggest to add a reference to WHO categories or refering in brackets something as “as defined in the Outcomes Measures section”.

Author’s response: thank you for nice suggestion. “as defined in the Outcome Measures section” was added in the manuscript. 

6. Line 169 (last line of the Blinding Table) states that “statistician” were not blind. Were these statisticians also involved in data results analysis? What was the role of this “unblind” statisticians? This is confusing

Author’s response: thank you. The confusing part is removed. They were not involved in data result analysis.

7. The authors explained in their response to reviewers that the placebo had the same excipients than colchicine pills, but in Line 184 the authors state that “placebo tasted similar to the study drugs”. Having the same excipients does not necessarily give the same bitter taste.

Author’s response: thank you. We do agree with you. The authors admitted that the bitter taste of colchicine could have compromised the effort to blind. Necessary changes were made at Line185-188 and 426-428. 

8. Line 264, add the year 2020 after December 15.

Author’s response: added

9. Table 1 shows results for 296 participants but 299 were randomized, this difference should be explained in the results section, line 265

Author’s response: thank you. The explanation is added at Line 268-270. 

10. Table 1, under comorbidities, several abbreviations, such as CLD, CKD, COPD are undefined. This was already mentioned by Reviewer nº2.

Author’s response: thank you. Correction is done. 

11. Was body mass index, or obesity measured in both groups? This is an important risk factor for severe COVID-19

Author’s response: thank you. Authors agree with you that body mass index (BMI) is an important risk factor but unfortunately there is no record of BMI in our case record form. 

12. Line 332, replace P-0.0035 with p=0.0035

Author’s response: thank you. Replaced

13. Line 331, remove upper-case from “Colchicine” and “Hazard” ratio

Author’s response: thank you. Removed

14. Line 335 “ALT” is not previously defined

Author’s response: thank you. Edited and defined when it first appeared at Table 1. 

15. Line 355-358 the description regarding liver testing abnormalities in both study arms is a little confusing, I suggest revising the writing

Author’s response: thank you. Edited at line 362 to 369. 

16. Discussion, line 379-380. The authors state that “colchicine-treated patients deteriorated 1 day later than the placebo group”. This finding was not statistically significant, therefore it should be removed.

Author’s response: thank you. The line is removed. 

17. Discussion: Line 383-389, revise text wording, several sentences are not well written

Author’s response: thank you. This part has been revised for word and language by Editage (www.editage.com). 

18. In general, the Discussion could be revised and much improved. For example, in line 396-402 all the discussion about CRP levels is unclear, I suggest revising carefully. How can the authors conclude that the CRP decrease relates to anti-inflammatory drug use and not just on clinical natural recovery of patients? This looks speculative.

Author’s response: thank you. Corrections have been made according to your suggestion (line 410-418). 

19. Any further discussion on the limitation of having over 2/3 of participants on both arms having received dexamethasone? Do the authors think that colchicine may have an additive beneficial effect? (this is where a multivariate analysis may have helped in the results interpretation…)

Author’s response: thank you. The authors agreed not to add further discussion. 

Reviewer #2: The authors have done well to address the comments made by reviewers. I believe the manuscript is a stage where readers will be able to judge well its merits and demerits. Regarding blinding, I will recommend to the authors to consider admitting that the bitter taste of colchicine could have compromised the effort to blind.

Author’s response: thank you for your nice recommendation. The authors admitted that the bitter taste of colchicine could have compromised the effort to blind. Necessary changes were made at Line184-186 and 411-413. 

Reviewer #3: Minor Revision:

Figure 3: What time points do the number at risk correspond to? The layout makes this difficult to determine. Placing the number at risk immediately below the Number of Days Survived on the x-axis may clarify.

Author’s response: thank you. It is corrected according to your suggestion. 

I hope I have tried my level best to address all of your concerns raised during the review process. Please consider my manuscript for publication in PLOS ONE.

Thanks

Dr. Ponkaj Kanti Datta

Assistant professor

Department of medicine

Dhaka Medical College

---

## [Decision Letter · Decision Letter 2]

29 Aug 2022

PONE-D-21-32054R2Efficacy of colchicine in patients with moderate COVID-19: A double-blinded, randomized, placebo-controlled trialPLOS ONE

Dear Dr. Datta,

Thank you for submitting your manuscript to PLOS ONE. After careful consideration, we feel that it has merit but does not fully meet PLOS ONE’s publication criteria as it currently stands.  We note that you have addressed the concerns raised by the reviewers, however, we recommend copyedit the newly added/corrected text for language usage, spelling, and grammar.  We suggest you to consult the editing company you have previously used to improve the above text.

We look forward to receiving your revised manuscript.

Kind regards,

Lorena Verduci

Staff Editor

PLOS ONE

Journal Requirements:

Reviewers' comments:

Reviewer's Responses to Questions

**Comments to the Author**

1. If the authors have adequately addressed your comments raised in a previous round of review and you feel that this manuscript is now acceptable for publication, you may indicate that here to bypass the “Comments to the Author” section, enter your conflict of interest statement in the “Confidential to Editor” section, and submit your "Accept" recommendation.

Reviewer #3: All comments have been addressed

2. Is the manuscript technically sound, and do the data support the conclusions?

Reviewer #3: (No Response)

3. Has the statistical analysis been performed appropriately and rigorously? 

Reviewer #3: (No Response)

4. Have the authors made all data underlying the findings in their manuscript fully available?

Reviewer #3: (No Response)

5. Is the manuscript presented in an intelligible fashion and written in standard English?

Reviewer #3: (No Response)

6. Review Comments to the Author

Reviewer #3: (No Response)

7. PLOS authors have the option to publish the peer review history of their article (what does this mean?). If published, this will include your full peer review and any attached files.

Reviewer #3: No

---

## [Author Response · Author response to Decision Letter 2]

10 Sep 2022

To 

Editor in chief,

PLOS ONE

Subject: In response to review of the manuscript entitled “Efficacy of colchicine in moderate COVID-19 patients: A double-blind, randomized, placebo controlled trial”.

Dear Sir,

Thank you for accepting our response to the concerns raised by the reviewer’s. We have copyedited the newly added/corrected text for language usage, spelling, and grammar by Editage as you recommended. I have reviewed the reference list carefully and checked the figure files with PACE to ensure that they meet PLOS requirements. 

I hope I have tried my level best to address all of your concerns raised during the review process. Please consider my manuscript for publication in PLOS ONE.

Thanks

Dr. Ponkaj Kanti Datta

Assistant professor

Department of medicine

Dhaka Medical College

---

## [Editor Report · Decision Letter 3]

4 Nov 2022

Efficacy of colchicine in patients with moderate COVID-19: A double-blinded, randomized, placebo-controlled trial

PONE-D-21-32054R3

Dear Dr. Datta,

We’re pleased to inform you that your manuscript has been judged scientifically suitable for publication and will be formally accepted for publication once it meets all outstanding technical requirements.

Kind regards,

James Mockridge

Staff Editor

PLOS ONE
---

## [Editor Report · Acceptance letter]

8 Nov 2022

PONE-D-21-32054R3 

Efficacy of colchicine in patients with moderate COVID-19: A double-blinded, randomized, placebo-controlled trial 

Dear Dr. Datta:

I'm pleased to inform you that your manuscript has been deemed suitable for publication in PLOS ONE. Congratulations! Your manuscript is now with our production department. 

Kind regards, 

on behalf of

Dr James Mockridge 

Staff Editor

PLOS ONE